# Contrasting genetic diversity between *Planchonella obovata* sensu lato (Sapotaceae) on old continental and young oceanic island populations in Japan

**Suzuki Setsuko**[1]*, **Kyoko Sugai**[2], **Ichiro Tamaki**[3], **Koji Takayama**[4], **Hidetoshi Kato**[5]

**1** Department of Forest Molecular Genetics and Biotechnology, Forestry and Forest Products Research Institute, Forest Research and Management Organization, Tsukuba, Ibaraki, Japan, **2** Institute of Agricultural and Life Sciences, Academic Assembly, Shimane University, Matsue, Shimane, Japan, **3** Gifu Academy of Forest Science and Culture, Mino, Gifu, Japan, **4** Department of Botany, Graduate School of Science, Kyoto University, Sakyo-ku, Kyoto, Japan, **5** Makino Herbarium, Tokyo Metropolitan University, Hachioji, Tokyo, Japan

* setsukos@affrc.go.jp

**Data Availability Statement:** Data for the SSR primers have been deposited in the DDBJ with

## Abstract

Genetic diversity of plant populations on islands is likely to be influenced by characteristics such as island origin (oceanic or continental) and their age, size, and distance to continental landmasses. In Japan, *Planchonella obovata* sensu lato which is found on both continental and oceanic islands of varying age, size, and distance to East Asian continental areas—is an ideal system in which to investigate the factors influencing genetic diversity of island plant species. In this study, we examined the genetic diversity of *P. obovata s.l.* populations, in the context of the species population genetic structure, demography, and between island migration, from 668 individuals, 28 populations and 14 islands including both continental (the Yaeyama Islands) and oceanic islands (the Daito, Bonin, and Volcano Islands) using 11 microsatellite markers. The Yaeyama and Volcano Islands respectively had the highest and lowest genetic diversity, and island origin and age significantly affected genetic diversity. Clustering analysis revealed that populations were grouped into Bonin, Volcano, and Yaeyama + Daito groups. However, Bonin and Volcano groups were distinct despite the relatively short geographical distance between them. Approximate Bayesian Computation analysis suggested that the population size was stable in Bonin and Yaeyama + Daito groups, whereas population reduction occurred in Volcano group, and migration between groups were very limited. Younger oceanic islands showed lower genetic diversity, probably due to limited gene flow and a lack of time to accumulate unique alleles. Genetic structure was generally consistent with the geographic pattern of the islands, but in Volcano, a limited number of founders and limited gene flow among islands are likely to have caused the large genetic divergence observed.

accession numbers LC076449- LC076466. Genotypic data of SSRs used for this study are available in S1 Data (Supporting Information). Data used in the lmer analysis to examine the associations between genetic diversity and island characteristics are available in S2 Data (Supporting Information).

**Funding:** "This work was funded by Grants-in-Aid for Science Research from the Japanese Society for Promotion of Science (18370038, 23310167, 26290073, 15K07203, 21K05694), the Environment Research and Technology Development Fund of the Ministry of the Environment, Japan (4-1402), and the support program of FFPRI for researchers having family responsibilities. The funder had no role in study design, data collection and analysis, decision to publish, or preparation of the manuscript."

**Competing interests:** The authors have declared that no competing interests exist.

## Introduction

Islands have long been important systems in ecology and evolutionary biology [1,2]. Due to the small size of their landmasses, isolation from source areas, simple biotas with relatively small number of species, and high levels of plant endemism, they provide excellent opportunities to investigate the evolutionary processes of plants. Islands are also suitable for studies of population genetics, to examine phenomena such as migration rates, degree of genetic isolation, and the extent of founder effects, since they are surrounded by water restricting the movement of terrestrial organisms [3]. The genetic diversity of plant populations on islands is likely to be influenced by island characteristics such as the geological origin of the island, its age, size, and distance to the nearest continental landmass. In the past decade, ecologists have applied island biogeography theory to investigations of genetic diversity, arguing that genetic and species diversity might be influenced by similar ecological processes [4]. For example, random extinctions of species in island communities are similar to the loss of alleles due to genetic drift [5], and thus factors influencing the species diversity of islands could influence the genetic diversity of the flora and fauna on islands. The genetic diversity of plant populations is generally lower on islands than in continental areas [6–8], and is lower on young islands than on old islands [9–11], although there are some exception [12,13]. Significant relationships have been found between genetic diversity and island area [14–17], and distance to the mainland [15,17–19]. Knowledge about genetic divergence within species, and its relationship to the island origin, age, size, and distance to the continent is essential to understanding the way in which plants colonize new islands and maintain genetic diversity.

Oceanic islands are defined as those which have never been connected to a continental landmass. They are the products of volcanism or tectonic uplift, or the results of organic reef growth upon foundations formed by the first two processes. Most continental islands were joined to other continental landmasses in the past, having since become separated due to tectonics or sea level rise [20]. In Japan there are oceanic and continental islands which have common subtropical climates, facilitating the comparison of genetic diversity between the islands. The Ogasawara Islands, including the Bonin and Volcano Islands, and the Daito Islands, are oceanic islands. However, the geneses of these islands differ, with the Ogasawara Islands being of volcanic origin while the Daito Islands are a result of uplifted atolls. The Yaeyama Islands are continental islands, which have been repeatedly connected to the Asiatic continent through Taiwan. The Bonin Islands are located 1,000 km south of mainland Japan in the Northwest Pacific Ocean, and include the Mukojima, Chichijima, and Hahajima Islands (Fig 1). They developed 44–34 mya [21,22], and were gradually uplifted before the middle Pleistocene [23]. The Volcano Islands are situated 150 km south of the Bonin Islands, and appeared 0.75–0.01 mya [24]. The Daito Islands, located about 360 km east of the mainland of Okinawa and about 1,000 km west of the Bonin Islands, are comprised of three small islands, Kitadaitojima, Minamidaitojima, and Okidaitojima. They developed on the sea bed 48 mya, sank under the sea 42 mya, and were uplifted again around 6 mya [25]. The Yaeyama Islands are located 400 km west of the mainland of Okinawa and 100 km east of the Taiwan. The South Ryukyu, where the Yaeyama Islands are located, developed between the late Paleozoic and the Mesozoic [26], and emerged around the Miocene (23 mya) [27,28]. Due to clear geological differences between these island groups, the genetic diversity, divergence and demography of plants are expected to differ between them.

*Planchonella obovata* sensu lato (Sapotaceae) is distributed in many parts of South-east Asia, south China, Taiwan, Micronesia, northeast Australia, and on islands of the Indian Ocean. In Japan, it occurs on the Ryukyu, Daito, Bonin, and Volcano Islands. *Planchonella obovata s.l.* is distributed widely from dry scrub to mesic forests on almost all of the Bonin

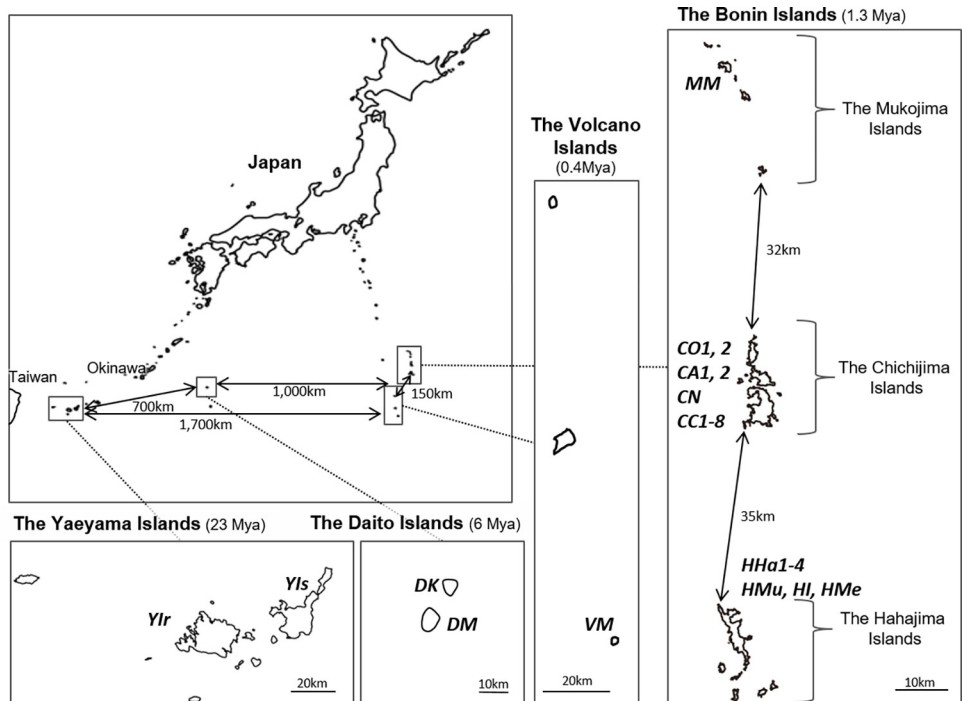

**Fig 1. Location of the Bonin, Volcano, Yaeyama and Daito Islands.** Bold italic letters show the Population ID of the sampled populations, shown in Table 1. Numbers in the parentheses are approximate age of islands used in the lmer analyses. Maps were drawn by ArcGIS using the coast line data downloaded from the Geographical Survey Institute of Japan (https://nlftp.mlit.go.jp/ksj/gml/datalist/KsjTmplt-C23.html), under a CC BY license, with permission from the Geographical Survey Institute of Japan, original copyright 2006.

Islands and from coastal areas to limestone areas in the mountains of the Ryukyu Islands, and is a major component of the natural vegetation. Thus, this species is ideal for the investigation of the genetic diversity, genetic structure, and population demography of plants in island areas of Japan.

The genus *Planchonella* includes around 110 species [29], and two species are found in Japan: *P. obovata s.l.* and *P. boninensis* (Nakai) Masam. et Yanagih. *Planchonella obovata s.l.* has two varieties, *P. obovata* (R.Br.) Pierre var. *obovata* and *P. obovata* var. *dubia* (Koidz. ex H. Hara) Hatus. ex T.Yamaz. *Planchonella obovata* var. *dubia* is distributed only on the Bonin and Daito Islands [30,31] where *P. obovata* var. *obovata* is also found. On these islands, *Planchonella obovata* var. *dubia*, inhabits dry rocky areas and is characterized by a lower stature and smaller leaves and fruits than *P. obovata* var. *obovata*, which grows in wetter habitats. However, the size of the leaves and fruits vary continuously between the two varieties, and are difficult to identify, and Ohashi and Kato [32] recognized *P. obovata* var. *dubia* as one of the ecotypes in dry areas.

In the Bonin Islands, adaptive radiation, probably caused by ecological divergence together with strong environmental gradients associated with the transition from mesic forests to dry scrub, has been observed in many genera such as *Callicarpa* [33], *Crepidiastrum*, *Pittosporum*, and *Symplocos* [34]. *Elaeocarpus photiniifolia*, endemic to the Bonin Islands, include genetically distinct groups associated with dry and mesic environments within the island [35]. Thus, the two varieties of *P. obovata* may be genetically distinct, although their phenotypic differences are slight.

The aim of this study was to answer following questions; (a) Are *P. obovata* var. *obovata* and *P. obovata* var. *dubia* genetically distinct? (b) What are the levels of population genetic

diversity of *P. obovata s.l.*, and which factors such as islands origin, age, area, and distance from the continent have the most impact on this diversity? (c) What is the *P. obovata s.l.* population structure in island areas in Japan? (d) Have the populations in each island group experienced past population size changes, such as expansion or reduction? (e) Is there migration among the island groups? The results of our study will contribute our understanding about colonization of continental plants on islands and how they maintain their genetic diversity.

## Materials and methods

### Study species and sample collection

*Planchonella obovata s.l.* is evergreen tree species, which is morphologically gynodioecious, although Kato et al. [36] have pointed out that it is functionally dioecious. Pollen is dispersed by insects, including flies, Oedemeridae beetle, and moths, which have been observed to visit flowers in the Bonin Islands [37,38]. Its berries are black, 1.2–1.5 cm long, and bear several 8–12 mm long seeds [30]. Primates, bats, lizards, and birds are thought to be important seed dispersers [39]. In the Bonin Islands, intact seeds are found in the feces of birds such as Japanese white-eye, Bonin Islands white-eye, and Brown-eared bulbul [40]. *Planchonella obovata s. l.* is distributed in both oceanic and continental islands, which vary in age, size, and distance to the continent, and is therefore appropriate for the investigation of the way in which genetic diversity is maintained in the islands.

We collected leaf samples of 663 individuals of *P. obovata s.l.* from 28 populations, 14 islands of the Bonin, Volcano, Daito, and Yaeyama Islands (Fig 1, Table 1). To ascertain whether *P. obovata* var. *obovata* and *P. obovata* var. *dubia* can be discriminated genetically, we sampled both where they occur in close proximity from the same location on Chichijima island (population CC4). These consisted of 17 individuals from dry rocky areas of typical *P. obovata* var. *dubia*, which is short in height with small leaves (CC4_dubia), and 35 individuals from wetter forest areas of typical *P. obovata* var. *obovata*, which is tall in height with large leaves (CC4). In the Daito Islands, there should be *P. obovata* var. *dubia* according to some previous reports, but we could not find typical *P. obovata* var. *dubia* there. After collection, the leaf samples were dried with silica gel. We recorded the locations of the sampled individuals using a GPS (GPSmap60CSx; Garmin, Olathe, Kansas, USA). Voucher specimens of each population were deposited in the herbarium of the Forestry and Forest Products Research Institute, Japan (nos. TF-FDA001379–TF-FDA001431) and Makino Herbarium, Tokyo Metropolitan University, Japan (nos. MAK378903, 378904, 391087).

### Microsatellite analysis

Total genomic DNAs of all sampled leaves were extracted, using DNeasy Plant Mini Kits (QIAGEN, Hilden, Germany). Eighteen nuclear microsatellite markers were developed by one *P. obovata* var. *obovata* plant from Hahajima in the Bonin Islands, and details of marker development was described in S1 Appendix and S1 Table in S1 File. We genotyped all 663 samples using 18 primer pairs, with the experimental conditions described in S1 Appendix in S1 File. We tested the existence of null alleles using Microchecker [41], and the linkage disequilibrium between loci in each population using FSTAT 2.9.3.2 [42].

Five (Po124, Po200, Po281, Po579, and Po583) out of 18 markers might have null alleles since estimated null allele frequency was significant in more than five populations. Two other markers, Po290 and Po623, were characterized by large amounts of missing data. No significant linkage disequilibrium was observed between loci in any population for the 11 markers, excluding the above mentioned seven markers. Thus, we used 11 markers for further population genetic analyses. Microsatellite genotype data for 11 markers are available in S1 Data.

**Table 1. Population genetic parameters estimated from 11 SSR for the 27 *Planchonella obovata* var. *obovata* and one *P. obovata* var. *dubia* population (CC4_dubia).**

| Island groups | Islands | Island area(ha) | Population ID | No. of samples | $A_R$ | $H_O$ | $H_E$ | $F_{IS}$ |
|---|---|---|---|---|---|---|---|---|
| The Bonin Islands | | | | | | | | |
| The Mukojima Islands | Mukojima | 256 | MM | 32 | 4.37 | 0.57 | 0.58 | 0.03 |
| The Chichijima Islands | Otoutojima | 520 | CO1 | 22 | 4.87 | 0.59 | 0.57 | 0.00 |
| | | | CO2 | 39 | 5.01 | 0.58 | 0.60 | 0.05 |
| | Anijima | 788 | CA1 | 44 | 4.68 | 0.57 | 0.59 | 0.05 |
| | | | CA2 | 29 | 4.84 | 0.55 | 0.59 | 0.09 |
| | Nishiijima | 48 | CN | 12 | 4.15 | 0.54 | 0.52 | 0.02 |
| | Chichijima | 2344 | CC1 | 20 | 4.79 | 0.56 | 0.57 | 0.05 |
| | | | CC2 | 21 | 4.36 | 0.52 | 0.55 | 0.09 |
| | | | CC3 | 17 | 4.50 | 0.50 | 0.53 | 0.09 |
| | | | CC4 | 35 | 4.88 | 0.54 | 0.58 | 0.09 |
| | | | CC4_dubia | 17 | 4.71 | 0.55 | 0.56 | 0.05 |
| | | | CC5 | 24 | 4.93 | 0.52 | 0.57 | 0.12 |
| | | | CC6 | 20 | 4.74 | 0.56 | 0.57 | 0.04 |
| | | | CC7 | 24 | 4.53 | 0.55 | 0.58 | 0.07 |
| | | | CC8 | 21 | 4.50 | 0.57 | 0.55 | 0.00 |
| The Hahajima Islands | Hahajima | 1988 | HHa1 | 23 | 4.54 | 0.56 | 0.59 | 0.07 |
| | | | HHa2 | 23 | 4.16 | 0.57 | 0.58 | 0.05 |
| | | | HHa3 | 23 | 4.35 | 0.59 | 0.59 | 0.03 |
| | | | HHa4 | 20 | 4.85 | 0.66 | 0.62 | -0.04 |
| | | | HHa5 | 25 | 4.38 | 0.62 | 0.62 | 0.03 |
| | Mukoujima | 138 | HMu | 20 | 4.71 | 0.56 | 0.62 | 0.12 |
| | Imoutojima | 123 | HI | 10 | 4.36 | 0.48 | 0.57 | 0.20 |
| | Meijima | 87 | HMe | 23 | 4.49 | 0.60 | 0.59 | 0.00 |
| The Volcano Islands | | | | | | | | |
| | Minamiiwoto | 354 | VM | 35 | 3.17 | 0.47 | 0.51 | 0.09 |
| The Daito Islands | | | | | | | | |
| | Kitadaitojima | 1194 | DK | 26 | 4.24 | 0.58 | 0.59 | 0.04 |
| | Minamidaitojima | 3057 | DM | 23 | 4.03 | 0.48 | 0.50 | 0.07 |
| The Yaeyama Islands | | | | | | | | |
| | Ishigakijima | 22250 | YIs | 14 | 5.28 | 0.58 | 0.65 | 0.13 |
| | Iriomotejima | 28930 | YIr | 21 | 5.69 | 0.72 | 0.72 | 0.03 |

$A_R$; allelic richness, $H_O$; observed heterozygosity, $H_E$; gene diversity, $F_{IS}$; fixation index.

All $F_{IS}$ values were not significantly deviated from Hardy-Weinberg equilibrium.

## Data analysis

**Statistical analyses of genetic diversity.** To evaluate the genetic diversity of each population, allelic richness ($A_R$) [43], observed heterozygosity ($H_O$) and gene diversity ($H_E$) [44] were calculated for each locus and each population using GenAlEx ver. 6.501 [45]. The fixation index ($F_{IS}$) was calculated and tested by randomization using FSTAT ver. 2.9.3 [46]. We used linear mixed-effect models (lmer) to examine the associations between genetic diversity ($A_R$, $H_E$) within each population and island characteristics such as its island age, area, origin (oceanic or continental), and distance to mainland China using the R package lme4 [47]. The island age (1: young; Volcano, 2: middle; Bonin and Daito, 3: old; Yaeyama), area (log-transformed), origin (1: oceanic, 0: continental), and distance to nearest continent were treated as

fixed effects, and the differences between loci as random effects. Island area was log-transformed to reduce skewness and increase the normality of its distribution. Variables were checked for collinearity using the variance inflation factor (VIF) value, which should be less than 5 [48], using the R package car [49]. The VIF for island origin was 5.11. We eliminated the island age from the lmer analysis, since island age and origin were highly correlated. We also conducted lmer analysis omitting the data of continental island populations, to eliminate the effect of island origin. The island age (0: young, 1: middle), area (log-transformed), and distance to nearest continent were treated as fixed effects, and the differences between loci as random effects. Data for $H_E$ were arcsine-transformed to obtain closer approximations to normality. Eight candidate models with 0–3 fixed explanatory variables were constructed for each response variable. We calculated Akaike's information criterion to evaluate the candidate models (AIC) [50] values for each of them. The differences between the AIC values and the minimum AIC value (ΔAIC) were calculated for each of the models, and models with ΔAIC value ≤ 2 were selected as the best models [51]. The analyses were conducted using R software 4.0.2 [52]. For the lmer analysis, data from individuals sampled as *P. obovata* var. *dubia* in population CC4_dubia were merged with population CC4, since there was no genetic difference between the two populations (see details in Results). Data used in the lmer analysis are available in S2 Data.

**Statistical analyses of genetic structure.** For population genetic structure analysis, we firstly conducted Bayesian clustering analysis using 35 typical *P. obovata* var. *obovata* and 17 typical *P. obovata* var. *dubia* individuals from the Bonin Islands, and 49 samples from the Daito Islands, to check whether *P. obovata* var. *dubia* was genetically different from *P. obovata* var. *obovata*, using the program STRUCTURE 2.3.4 [53,54]. In this analysis, we chose allele frequency correlated model and admixture model to detect the admixture of lineages, and each run involved 100,000 Markov chain Monte Carlo (MCMC) iterations after a burn-in period of 50,000 iterations. The analysis was run 30 times with each *K*, ranging from 1 to 10. Then, the population genetic structure of all *P. obovata s.l.* populations were investigated using STRUCTURE, reducing the sample size of the Chichijima and Hahajima Islands populations to 45 randomly selected individuals each, since the program may show a bias when the sampling design is unbalanced [55]. In this analysis, each run involved 100,000 MCMC iterations after a burn-in period of 50,000 iterations. The analysis was run 30 times with each *K*, ranging from 1 to 15. The optimal value for *K* was evaluated using the Δ*K* [56] and the mean log likelihood at each *K* [54]. STRUCTURE tends to detect the highest level of a population hierarchy [56], and there can be lower hierarchy of structuring within the highest clusters. Thus, we also explored the substructure within each detected cluster. Results were summarized using the CLUMPAK [57].

The genetic relationships among populations were assessed by a neighbor-joining (NJ) tree based on the $D_A$ genetic distance [46] between them, using the program Populations 1.2.32 [58]. The significance of each node in the tree was evaluated by 1,000 bootstraps. The isolation by distance (IBD) [59] pattern was evaluated by the Mantel test [60] on population pairwise natural logarithms of geographical distance (ln (1 + geographical distance)) and $F_{ST}/(1 - F_{ST})$ [61].

**Statistical analyses of population demography.** STRUCTURE analysis detected clear genetic structure among three island groups: the Bonin Islands; the Volcano Islands; and the Yaeyama and Daito Islands (see details in Results). In order to estimate the population demography of these three island groups, we conducted approximate Bayesian computation (ABC). As there are various patterns in combinations of population demography—population size change, population divergence, and migration patterns—we sequentially executed ABC analyses [11,62]. In the first step, we applied single population size change models for each island

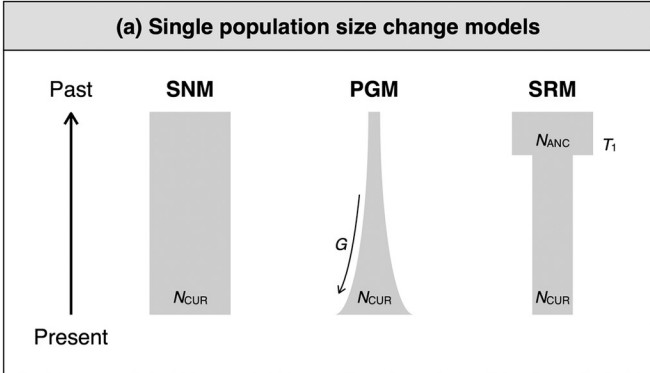
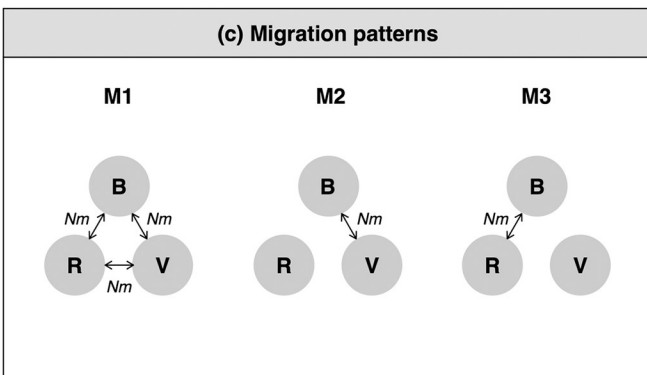
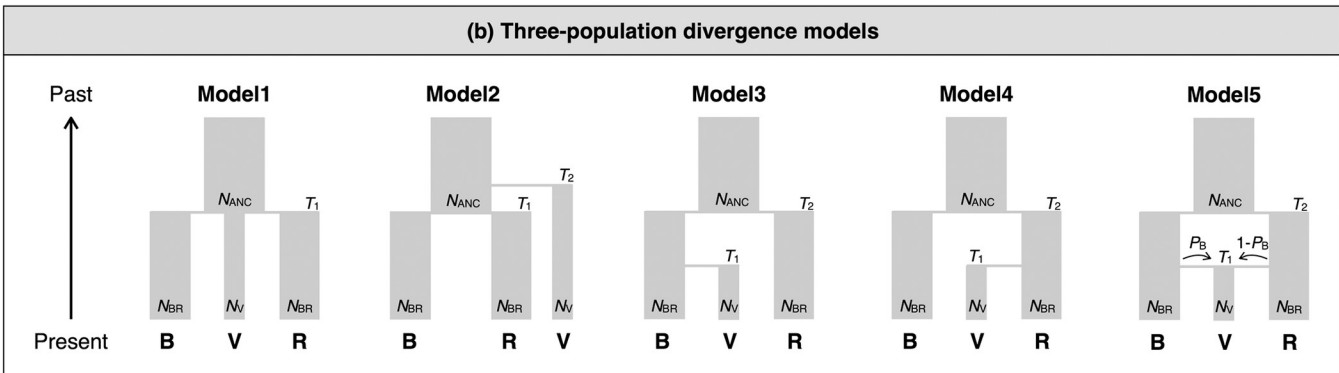

**Fig 2. Population demographic models.** Three single population size change models (a; SNM, standard neutral model; PGM, population growth model; SRM, size reduction model). Five three-population divergence models (b; B, Bonin group; V, Volcano group; Y+D, Yaeyama and Daito group). Three migration patterns (c).

group (Fig 2A). In the second step, we applied three-population divergence models without migration for the three island groups (Fig 2B). Finally, in the third step, we examined divergence models with and without migration (Fig 2C). The detail of ABC analysis was described in S2 Appendix in S1 File.

## Results

### Genetic difference between *P. obovata* var. *obovata* and *P. obovata* var. *dubia*

According to a NJ tree based on the $D_A$ genetic distance, the *P. obovata* var. *obovata* (CC4) and *P. obovata* var. *dubia* (CC4_dubia) sampled from the same site on Chichijima island in the Bonin Islands, formed a cluster with a high bootstrap value (Fig 3). The pairwise $F_{ST}$ between CC4 and CC4_dubia was low as 0.012 (Fig 4). STRUCTURE analysis showed no genetic differentiation between CC4 and CC4_dubia with increasing $K$ (S1 Fig in S1 File). In the Daito Islands, there was genetic differentiation between the two sampled populations, Kita-daitojima and Minamidaitojima, however no clear genetic sub-structuring which would imply the existence of *P. obovata* var. *dubia* was found within populations.

### Genetic diversity

At the 11 SSR loci examined, allelic richness ($A_R$) ranged from 3.17–5.69 (mean 4.58), and gene diversity ($H_E$) ranged from 0.50–0.72 (mean 0.58, Table 1). The $A_R$ values of each island

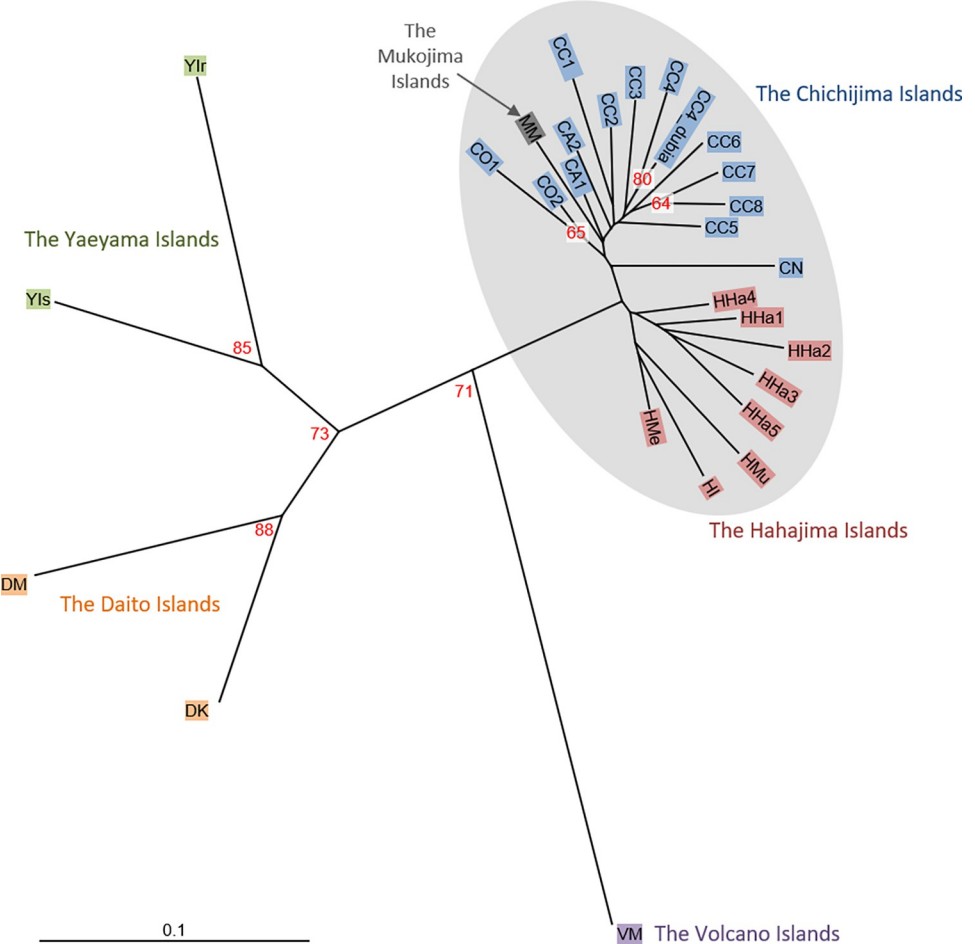

**Fig 3. A neighbor-joining dendrogram for 27 *Planchonella obovata* var. *obovata* and one *P. obovata* var. *dubia*
populations based on 11 SSR markers based on genetic distance, $D_A$.** Numbers in the internal nodes indicate
bootstrap value larger than 50; gray oval, populations in the Bonin Islands.

group were significantly different (Kruskal–Wallis test, $p < 0.01$). The $A_R$ values of the
Yaeyama Islands were significantly higher than that of the Volcano Islands (pairwise t-test,
$p < 0.01$, Fig 5A), and those of the Yaeyama and Daito, and Bonin and Volcano Islands were
marginally significant (pairwise t-test, $p = 0.08, 0.07$, respectively). The $H_E$ values of each island
group were significantly different (Kruskal–Wallis test, $p < 0.05$), and the $H_E$ of the Yaeyama
and Volcano islands was marginally significant (pairwise t-test, $p = 0.08$, Fig 5B).

In the lmer analysis investigating the associations between genetic diversity and island char-
acteristics for all populations, we eliminated island age from the analysis, since island origin
and age were highly correlated. For the lmer analysis explaining $A_R$ and $H_E$, the best model
both consisted only of island origin (S2 Table in S1 File), and the coefficients of island origin
were negative for both $A_R$ and $H_E$ (S3 Table in S1 File). To eliminate the effect of island origin,
we also conducted lmer analysis using only oceanic islands data. For the analysis explaining
$A_R$, the best model included only island age. For the analysis explaining $H_E$, the null model had
the smallest AIC, while the Δtheilet AICsecond-best model was ≤ 2, and included island age.
Coefficients of island age were positive for both $A_R$ and $H_E$.

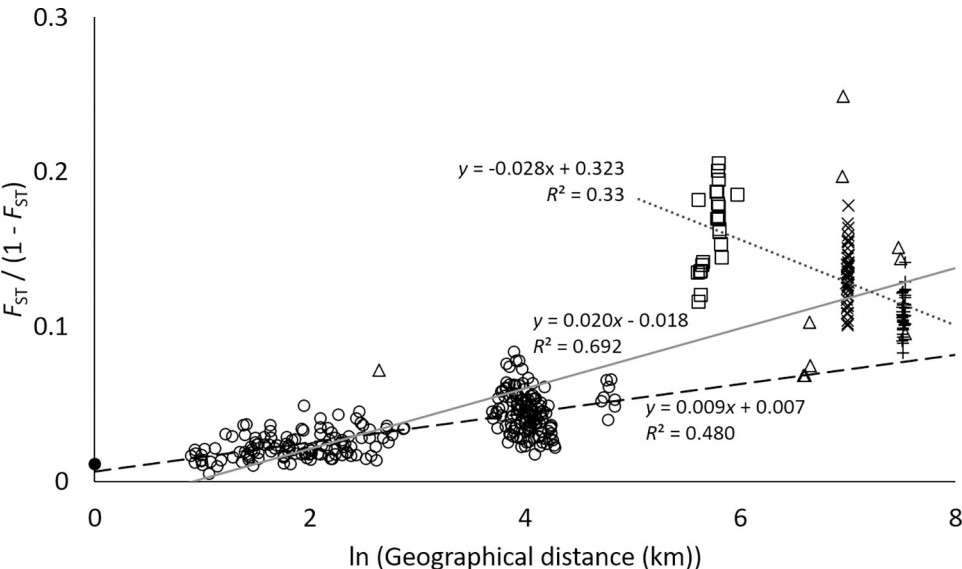

**Fig 4. Relationship between pairwise genetic differentiation and natural logarithms of geographical distance for all the 27 *P. obovata* var. *obovata* and one *P. obovata* var. *dubia* populations.** Solid circle, population pair between *P. obovata* var. *obovata* (CC4) and *P. obovata* var. *dubia* (CC4_dubia); open circles, population pairs within the Bonin Islands; squares, population pairs between the Bonin and Volcano Islands; cross signs, population pairs between the Bonin and Daito Islands; plus signs, population pairs between the Bonin and Yaeyama Islands; triangles, population pairs between other islands; solid gray line, regression line for all 28 populations; black broken line, regression line for population pairs within the Bonin Islands; gray dotted line, regression line for population pairs whose geographical distance over 270 km (ln (1 + geographical distance > 5.6)).

## Genetic structure

In an NJ tree based on genetic distance, $D_A$, *P. obovata s.l.* populations clustered into four distinct groups: the Bonin, Volcano, Daito, and Yaeyama Islands with high bootstrap values (Fig 3). In the Bonin Islands, populations were grouped into two: the Mukojima and Chichijima Islands, and the Hahajima Islands. The Daito Islands were closer to the Yaeyama Islands than to the Bonin Islands. The Volcano Islands population was located between the Bonin Islands and the Yaeyama and Daito Islands, with a long branch.

STRUCTURE analysis for all *P. obovata s.l.* populations showed that the $\Delta K$ was highest when $K = 3$: the Bonin Islands; the Volcano Islands; and the Yaeyama and Daito Islands were separated (Fig 6). At $K = 4$, the Daito and Yaeyama islands were separated. At $K = 5$, the Mukojima (MM) and Chihijima Islands (CO1-CC8), and the Hahajima Islands (HHa1-HMe). At $K = 6$, the Mukojima and Chihijima Islands were separated. At $K = 7$, the log likelihood reached highest value, Kitadaito (DK) and Minamidaito Islands (DM) were separated.

There were significant IBD among all populations ($R^2 = 0.692$, $p < 0.05$, Fig 4) and among populations in the Bonin Islands ($R^2 = 0.480$, $p < 0.01$), and regression coefficients were higher among all populations (all populations = 0.020; the Bonin Islands = 0.009). On the other hand, significant negative correlation was detected between geographical distance and pairwise $F_{ST}$ among populations whose geographical distance over 270 km (ln (1 + geographical distance) > 5.6, $r = -0.577$, $p < 0.001$). Higher pairwise $F_{ST}$ values were observed between the populations of the Bonin and Volcano Islands.

## Population demography and divergence

The best of the three single population size change models, as selected by ABC-RF, were the standard neutral model (SNM) in the Bonin and Yaeyama + Daito groups, and the size

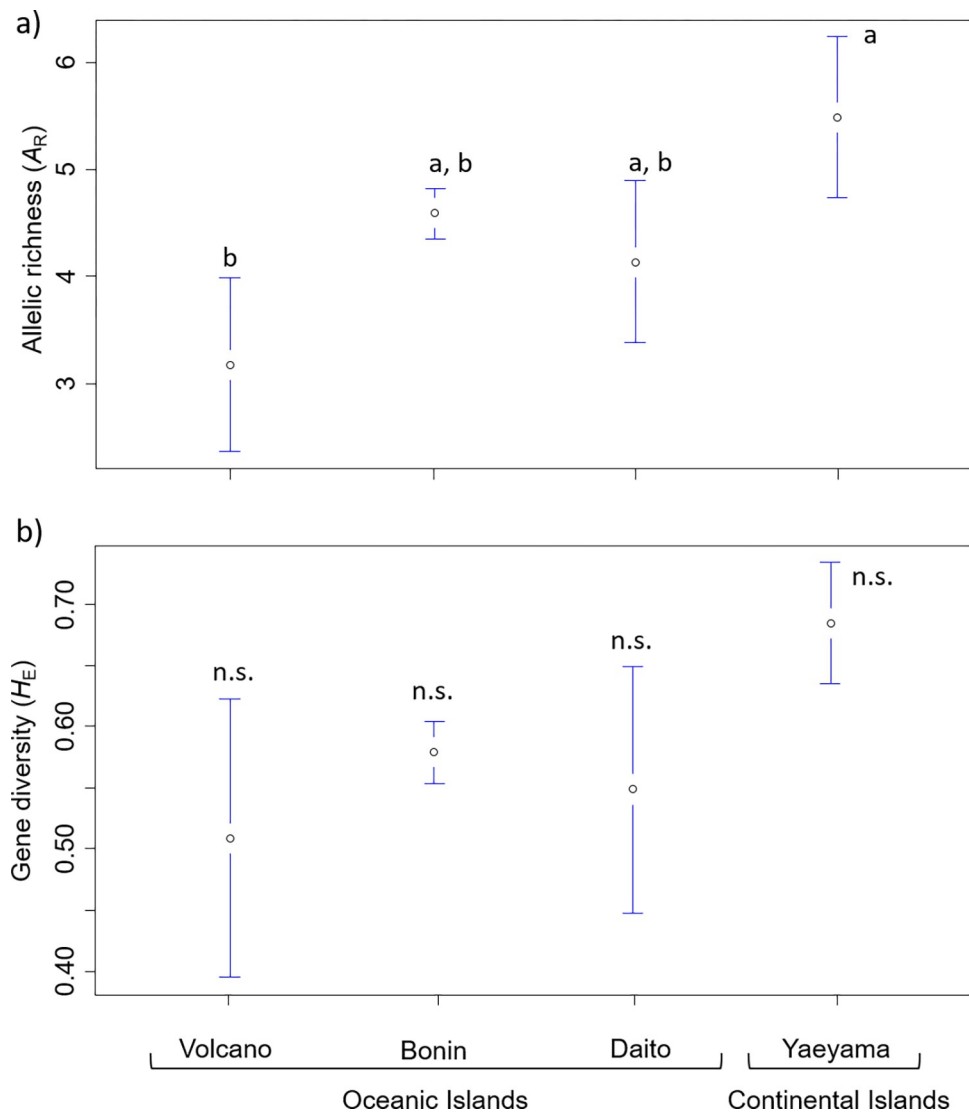

**Fig 5. Mean allelic richness ($A_R$) and gene diversity ($H_E$) in the four islands of *Planchonella ovobata* (± SE).**
Different letters indicate significant differences among islands ($p < 0.05$, pairwise t-test with Bonferroni correction).
Island groups are ordered from youngest (left) to oldest (right).

reduction model (SRM) in the Volcano group, with relatively low error rates (0.255–0.279) and high posterior probabilities (0.757–0.791; Fig 2A, S4 Table in S1 File). Using the best models, we estimated the posterior distribution of parameters. All parameters, except for the relative ancestral effective population size ($RN_{ANC}$) in $\log_{10}$ scale, and the mean geometric parameter in the generalized stepwise mutation model ($P_{GSM}$) in the SRM of the Volcano Island group, showed a clear single peak (S2 Fig in S1 File). The Bonin and Yaeyama + Daito groups showed similar level of current effective population size ($N_{CUR}$) and its posterior modes (95% HPD) were 10,436 (6,517–18,411) and 12,468 (7,283–23,472), respectively, while Volcano had a smaller value of $N_{CUR}$ than the others, at 1,829 (401–4,561; S4 Table in S1 File). Most of the summary statistics predicted by the posterior distribution fell near the observed values, and we concluded that the goodness-of-fit values of the single population size change models to the observed data were high (S3 Fig in S1 File).

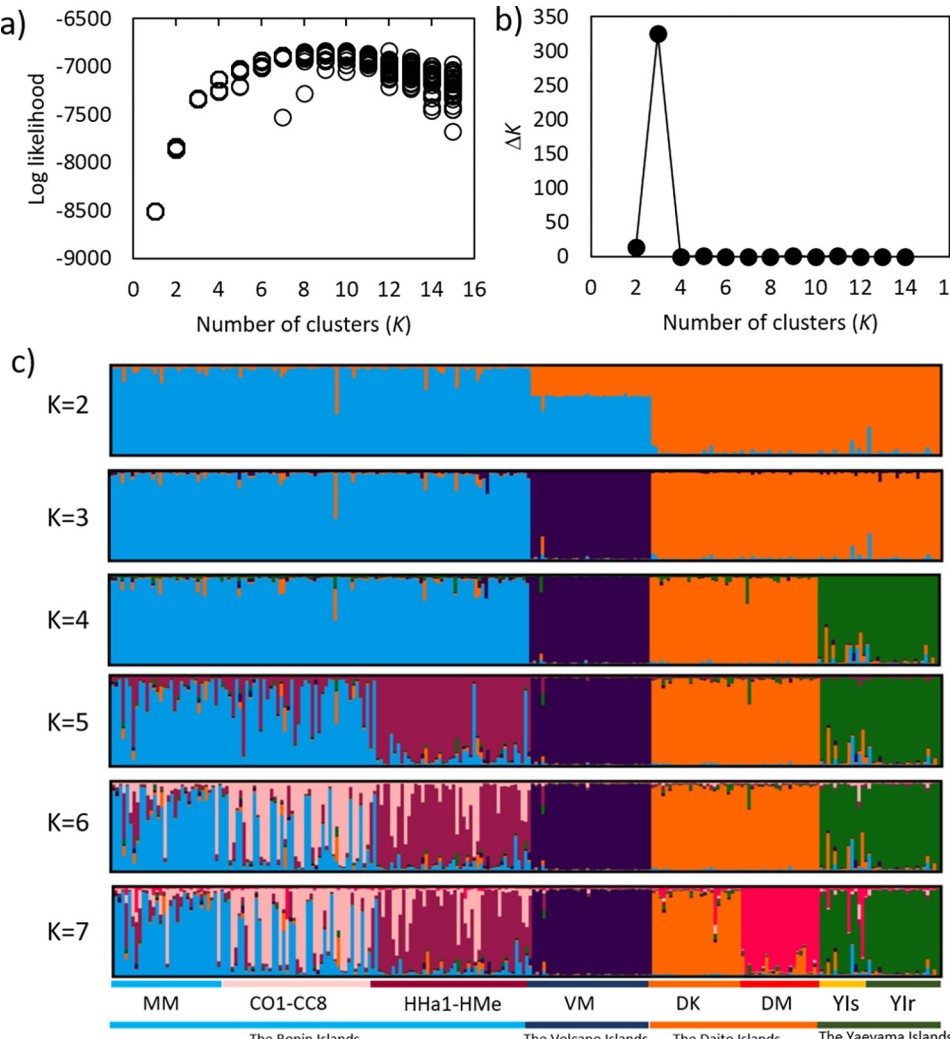

**Fig 6. Results of STRUCTURE analysis for all *Planchonella obovata s.l.* populations based on 11 SSR markers.**
Forty-five samples from the Chichijima and Hahajima Islands were selected at random. Changes in loglikelihood and
ΔK as the number of clusters (K ranging from 1 to 15), barplots of 241 genotypes at K = 2 to 9. Vertical columns
represent individuals; heights of bars are proportional to the posterior means of the estimated admixture proportions.

In a comparison of the five three-population divergence models, Model3 was the best
model, with a posterior probability of 0.431 (Fig 2B, Table 2). However, votes of Model1
(0.265) was not low compared with those of Model3 (0.369), and the error rate was also rela-
tively high (0.388). This high error rate was due to the high classification error rate of Model5,
as shown in the confusion matrix (0.646; S5 Table in S1 File). Thus, we removed Model5 and
compared the remaining four models. In this comparison, although Model3 was again selected
as the best model, and the error rate was reduced (0.295), the posterior probability was still not
high (0.516), and the votes of Model1 were also still not low (0.352) compared to those of
Model3 (0.429) (Table 2). As the votes for the other two models were low (0.117 for Model2
and 0.102 for Model4), we removed these two models and compared only Model1 and
Model3. In this comparison, Model3 was again selected as the best model, and had a high pos-
terior probability (0.752) and a low error rate (0.184) (Table 2). Therefore, we confirmed that
Model3 was the best model among the five three-population divergence models. Using the

**Table 2. Proportion of votes by random forest, posterior probability of the best model, and classification error rate in population divergence and migration analyses.**

| Compared model set | Proportion of votes [a] | | | | | Posterior probability | Classification error rate |
|---|---|---|---|---|---|---|---|
| | Model1 | Model2 | Model3 | Model4 | Model5 | | |
| Five divergence models | 0.265 | 0.096 | **0.369** | 0.068 | 0.202 | 0.431 | 0.388 |
| Four divergence models | 0.352 | 0.117 | **0.429** | 0.102 | — | 0.516 | 0.295 |
| Two divergence models | 0.366 | — | **0.634** | — | — | 0.752 | 0.184 |
| | Model3M1 | Model3M2 | Model3M3 | Model3 | | | |
| With/without migration models | 0 | 0.004 | 0.018 | **0.978** | | 0.954 | 0.216 |

[a] Best model is shown in bold.

best divergence model, Model3, we compared models with and without migration using ABC-RF, and the without migration model was strongly supported, having a posterior probability of 0.954 (Fig 2C, Table 2). We thus estimated the posterior distribution of the parameters in Model3, and all parameters showed a clear single peak (S4 Fig in S1 File). The posterior mode (95% HPD) of the current effective population sizes in the Bonin and Yaeyama + Daito groups ($N_{BYD}$), and the Volcano group ($N_V$) were 15,213 (8,743–29,273) and 1,710 (690–4,537), respectively (S6 Table in S1 File). The posterior mode (95% HPD) of the time of divergence of the Volcano group from the Bonin group ($T_1$) was 1,623 (295–5,169) generations ago. That between the Bonin and the Yaeyama + Daito groups ($T_2$) was 8,652 (2,888–21,509) generations ago. Most of the summary statistics predicted by the posterior distribution fell near the observed values, and we concluded that the goodness-of-fit of Model3 to the observed data was high (S5 Fig in S1 File).

## Discussion

### *P. obovata* var. *obovata* and *P. obovata* var. *dubia*

We could not differentiate *P. obovata* var. *obovata* and *P. obovata* var. *dubia* samples collected in the Bonin Islands genetically using either a NJ tree or STRUCTURE analysis. In the Daito Islands, no genetic sub-structuring which would imply the existence of *P. obovata* var. *dubia* were found within populations using the STRUCTURE analysis. Thus, we concluded that *P. obovata* var. *dubia* is one of the phenotypic variations of *P. obovata* var. *obovata* in this study. However, considering that the leaves of the individuals we sampled from the Daito Islands were larger than those of specimens identified as *P. obovata* var. *dubia* in the Daito Islands (nos. RYU4937 and RYU4955), there is a possibility we did not sample genuine *P. obovata* var. *dubia* there. More intensive search for typical *P. obovata* var. *dubia* and research combining genetic data and phenotypic data such as the sizes of leaves and fruits should be conducted in the Daito Islands in the future.

### Genetic diversity of *Planchonella obovata* sensu lato

Allelic richness ($A_R$) was highest in the Yaeyama Islands (mean 5.49), moderate in the Bonin and Daito Islands (means 4.60 and 4.14, respectively), and lowest in the Volcano Islands (3.17). Gene diversity ($H_E$) showed the same pattern as $A_R$, being highest in the Yaeyama Islands (mean 0.68), moderate in the Bonin and Daito Islands (means 0.58 and 0.55, respectively), and lowest in the Volcano Islands (0.51). This is consistent with previous studies with Maki [63] finding that allozyme diversity of plants endemic to the continental Ryukyu Islands ($H_T$ 0.134–0.321) are about five times higher than plants endemic to the oceanic Bonin Islands

($H_T$ 0.000–0.0083). In addition, using microsatellites, lower genetic diversity in the Volcano Islands compared with the Bonin Islands were also observed in *Pandanus boninensis*, an endemic species in the Bonin and Volcano Islands [11].

The lmer analyses suggested that island origin and age contributed most to the $A_R$ and $H_E$ levels. Island origin affected the genetic diversity, with populations in the continental islands having higher genetic diversity than oceanic ones. This is probably due to difference in number of founders with populations on oceanic-islands being established from limited seeds arriving via rare long-distance dispersal. In contrast, populations on continental-islands should have experienced greater seed influx during and post- establishment through seed dispersal by birds and other animals in the past when the continental islands were connected to continental landmasses. Island age also affected the genetic diversity, with populations on older islands having higher genetic diversity. This observation is consistent with studies in other island groups, such as the Canary Islands [9] and New Caledonia [10]. This phenomenon probably arises because unique alleles have not had enough time to accumulate via mutation and recombination on younger islands [64,65]. However, island area and distance to nearest continent did not contribute to the observed differences in genetic diversity. Although, higher genetic diversity in large populations is usually observed [66], and in the Bonin Islands, a positive correlation between genetic diversity and island area has been found in *Pa. boninensis* [11], in *P. obovata s.l.*, island size may not be a good indicator of population size since its distribution is restricted to the coastal area especially in the Yaeyama Islands. Interestingly, distance to nearest continent did not impact genetic diversity probably because geographical distance used in lmer were too large to test the effect on the genetic diversity of *P. obovata s.l.* This is likely due to the fact that migration among island groups is extremely low, as evidenced by Model3 without migration being selected in the population divergence analysis, and, although overall isolation by distance was significant, a significant negative correlation for the population pairs whose distance over 270 km was observed, suggesting no isolation by distance pattern over this geographical distance. Given that the shortest distance to nearest continent in our data is 450 km (the Yaeyama Islands) recent migration into investigated islands from continental areas would also very limited and might have no effects on the genetic diversity.

**Genetic structure of *Planchonella obovata* sensu lato.**   NJ tree and STRUCTURE analysis of all *P. obovata s.l.* populations identified three genetic groups: the Bonin Islands; the Volcano Islands; and the Yaeyama and Daito Islands. This finding was broadly consistent with the geographic arrangement of the islands. The Yaeyama and Daito Islands populations were adjacent in the NJ tree, and they were in the same cluster in the STRUCTURE analysis when $K = 4$, suggesting that the younger Daito Islands populations may have diverged from the older Ryukyu Islands where the Yaeyama Islands are located. This result is plausible, because most of the plants in the Daito Islands are shared with the Ryukyu Islands [67]. The Volcano Islands population was equally distant from the Bonin and Yaeyama + Daito Islands populations in the NJ tree, and STRUCTURE analysis could not determine which genetic groups were close to the Volcano Islands (Fig 6C, $K = 2, 3$). Population divergence analysis was able to unravel this question. Model3, in which the Bonin and Yaeyama + Daito groups diverged from the ancestral population, and then the Volcano group diverged from the Bonin group, was selected as the best model. The younger Volcano Islands population was inferred as having diverged from the older Bonin Islands populations. This pattern was also found in *Pa. boninensis* in the same area [11], and other islands, *Plantago* in the Hawai'ian Islands [68] and *Drimys* in the Juan Fernandez archipelago [69]. We only sampled *P. obovata s.l.* in Japan, and other candidates, such as Northern Mariana Islands, 540 km south of the Volcano Islands, were not included in our analyses. However, if the Mariana Islands is a source population of the Volcano Islands,

Model1 or 2 should have been selected by ABC-RF. We should therefore include these islands to correct the estimation of the origins of the populations in the future.

The results of NJ tree, STRUCTURE analysis, and IBD analysis indicated populations in the Bonin and Volcano Islands were more distinct than may be expected given the relatively short geographical distance between them. This is probably caused by the following two factors. One is the founder effect would have occurred in the Volcano Islands since the size reduction model (SRM) was selected from population size change analysis in the Volcano Island group. This finding suggests that the number of founders which migrated from the Bonin Islands to the Volcano Islands was very limited. The other is gene flow from the Bonin Islands to the Volcano Islands was very limited as described above, and thus increased the genetic divergence between them.

In the Bonin Islands, *P. obovata s.l.* populations were clustered into two genetic groups: the Mukojima and Chichijima Islands, and the Hahajima Islands using NJ tree and STRUCTURE analysis with $K = 5$. This genetic pattern is also found in other flora and fauna dispersed by ocean currents in the Bonin Islands, such as *Terminalia catappa* [70], *Pa. boninensis* [11], *Hibiscus* [71], lizards [72] and land snails [73]. Fruits of *P. obovata s.l.* consist of a black berry and are thought to be adapted to dispersal by birds. Intact seeds of *P. obovata s.l.* were found in feces of Japanese white-eyes, Bonin Islands white-eyes, and Brown-eared bulbuls in the Bonin Islands [40]. Brown-eared bulbuls have high mobility, however, they are unlikely to disperse seeds between the Mukojima and Chichijima Islands, which are separated by 32 km, or the Chichijima and Hahajima Islands, which are 35 km apart, according to the retention time in the guts of the Brown-eared Bulbul, which is 30 min at most for 9.3 mm seeds [74] and their cruising speed of 29–36 km/h, calculated from body weight [75]. However, Swenson *et al.* [76] reported that *Planchonella* was dispersed as far as 8,900 km between Palau in the Pacific and the Seychelles in the Indian Ocean, based on a maximum clade credibility tree. Seeds of *P. obovata s.l.* are resistant to sea water [77]. Similar genetic patterns have been observed in ocean-distributed species, which have the potential to be dispersed over very long distances by water, and the salt tolerance of the seeds suggests that seeds would be dispersed by ocean currents among the Mukojima, Chichijima, and Hahajima Islands. This phenomenon whereby seeds are dispersed by vectors different from those to which they are best suited is called non-standard mechanisms of dispersal [78], and could play a role in the long distance dispersal of this species, resulting in island colonization [79].

## Population demography and divergence times

In the population size change analysis, the SNM was selected in the Bonin, Yaeyama + Daito groups, and the SRM was selected in the Volcano group. These results suggest that the population size was stable in the Bonin, Yaeyama and Daito Islands, while population reduction occurred in the Volcano Islands. A similar analysis was conducted for *Pa. boninensis* in the Bonin and Volcano Islands, and the population growth model (PGM) was selected in the Bonin Islands, while the SRM was selected for the Volcano Islands [11]. The SRM was selected in the Volcano Islands for both species, suggesting that the effect of a founder event in the young islands is still detectable. The divergence time of the Bonin and Yaeyama + Daito groups from the ancestral population ($T_2$) for *P. obovata s.l.* was 8,652 generations ago, while that of Bonin from the ancestral population for *Pa. boninensis* was 91,925 (S4 Table in S1 File in Setsuko *et al.* [11]), and *P. obovata s.l.* is one digit younger than *Pa. boninenisis*, assuming that the generation time of *P. obovata s.l.* and *Pa. boninensis* is almost the same. This finding is consistent with the fact that the *Pa. boninensis* is endemic to the Bonin and Volcano Islands, so a sufficient amount of time has passed for the ancestral *Pandanus* to have speciated into an

endemic species. In contrast, the colonization of *P. obovata s.l.* in the Bonin Islands is likely to be relatively recent, and colonization of the islands already occupied by other plant species, may prevented the species from undergoing an increase in population size as *Pa. boninenisis* did, and thus the SNM would be selected in the population size change analysis. However, population demography and divergence time estimates of *P. obovata s.l.* in this study were derived from only limited number of SSR markers using simple ABC approaches, and further investigation using markers with higher resolution such as genome wide SNPs and other statistical approaches should be undertaken in the future.

## Conclusion

We examined the genetic diversity, structure, and population demography of *P. obovata s.l.* on both continental (the Yaeyama Islands) and oceanic islands (the Daito, Bonin, and Volcano Islands) using 11 microsatellite markers. We could not differentiate *P. obovata* var. *obovata* and *P. obovata* var. *dubia* genetically, and concluded that *P. obovata* var. *dubia* is part of the phenotypic variation found in *P. obovata* var. *obovata*. Island origin and age had significant effects on the genetic diversity of *P. obovata s.l.* Genetic diversity was higher in the old continental islands (the Yaeyama Islands) and lower in the young oceanic islands (The Volcano Islands). This difference was probably caused by two reasons. One is difference in number of founders, which is greater on continental islands, and the other is difference of time to accumulate the new alleles by mutation and recombination. Genetic structure was generally consistent with the geographic pattern of the islands, but in the young oceanic islands, a limited number of founders and limited gene flow among islands is likely to have caused the large genetic divergence observed. ABC analysis revealed population size was stable in the old continental and older oceanic islands (the Bonin Islands), while population reduction occurred in the young oceanic islands, migration among the island groups were very limited, and suggested that the young oceanic islands were colonized by geographically close, older oceanic islands. Results of our study provides a good example about colonization of continental plants on islands and how they maintain their genetic diversity.

## Supporting information

**S1 File.**
(DOCX)

**S1 Data. Microsatellite genotype data for 11 markers.**
(XLSX)

**S2 Data. Data used in the lmer analysis.**
(XLSX)

## Acknowledgments

The authors are grateful to Dr. K. Kawakami and Dr. J.R.P. Worth of Forestry and Forest Products Research Institute for their valuable advice; Dr. T. Denda and Dr. M. Yokota of University of the Ryukyus to provide us voucher specimen data of *Planchonella* in the Daito Islands; A. Hisamatsu and Y. Kawamata for their experimental support. We also thank Metropolis of Tokyo, the Ministry of the Environmental Government of Japan and Forestry Agency of Japan for allowing this study. This research was conducted using the Ogasawara Field Research Station of Tokyo Metropolitan University.

## Author Contributions

**Conceptualization:** Suzuki Setsuko, Kyoko Sugai, Hidetoshi Kato.

**Data curation:** Suzuki Setsuko, Kyoko Sugai, Koji Takayama, Hidetoshi Kato.

**Formal analysis:** Suzuki Setsuko, Ichiro Tamaki.

**Funding acquisition:** Suzuki Setsuko, Hidetoshi Kato.

**Investigation:** Suzuki Setsuko, Kyoko Sugai, Koji Takayama, Hidetoshi Kato.

**Methodology:** Suzuki Setsuko, Ichiro Tamaki.

**Project administration:** Suzuki Setsuko, Hidetoshi Kato.

**Software:** Ichiro Tamaki.

**Supervision:** Suzuki Setsuko.

**Visualization:** Suzuki Setsuko, Ichiro Tamaki.

**Writing – original draft:** Suzuki Setsuko, Ichiro Tamaki.

**Writing – review & editing:** Suzuki Setsuko, Ichiro Tamaki.

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
