## [Decision Letter · Decision Letter 0]

18 Apr 2022

PONE-D-22-03882Contrasting genetic diversity between Planchonella obovata sensu lato (Sapotaceae) on old continental and young oceanic island populations in JapanPLOS ONE

Dear Dr. Setsuko,

Thank you for submitting your manuscript to PLOS ONE. After careful consideration, we feel that it has merit but does not fully meet PLOS ONE’s publication criteria as it currently stands. Therefore, we invite you to submit a revised version of the manuscript that addresses the points raised during the review process.

We look forward to receiving your revised manuscript.

Kind regards,

Ricardo Alia, PhD

Academic Editor

PLOS ONE

Journal Requirements:

“HK; 18370038; Grants-in-Aid for Science Research from the Japanese Society for Promotion of Science

HK; 23310167; Grants-in-Aid for Science Research from the Japanese Society for Promotion of Science

HK; 26290073; Grants-in-Aid for Science Research from the Japanese Society for Promotion of Science

SS; 15K07203; Grants-in-Aid for Science Research from the Japanese Society for Promotion of Science

SS; 21K05694; Grants-in-Aid for Science Research from the Japanese Society for Promotion of Science

SS; 4-1402; the Environment Research and Technology Development Fund of the Ministry of the Environment, Japan

SS; 95200; the support program of FFPRI for researchers having family responsibilities”

“This work was funded by Grants-in-Aid for Science Research from the Japanese Society for Promotion of Science (18370038, 23310167, 26290073, 15K07203, 21K05694), the Environment Research and Technology Development Fund of the Ministry of the Environment, Japan (4-1402), and the support program of FFPRI for researchers having family responsibilities.”

“HK; 18370038; Grants-in-Aid for Science Research from the Japanese Society for Promotion of Science

HK; 23310167; Grants-in-Aid for Science Research from the Japanese Society for Promotion of Science

HK; 26290073; Grants-in-Aid for Science Research from the Japanese Society for Promotion of Science

SS; 15K07203; Grants-in-Aid for Science Research from the Japanese Society for Promotion of Science

SS; 21K05694; Grants-in-Aid for Science Research from the Japanese Society for Promotion of Science

SS; 4-1402; the Environment Research and Technology Development Fund of the Ministry of the Environment, Japan

SS; 95200; the support program of FFPRI for researchers having family responsibilities”

5. We note that Figure 1 in your submission contain map images which may be copyrighted. All PLOS content is published under the Creative Commons Attribution License (CC BY 4.0), which means that the manuscript, images, and Supporting Information files will be freely available online, and any third party is permitted to access, download, copy, distribute, and use these materials in any way, even commercially, with proper attribution. For these reasons, we cannot publish previously copyrighted maps or satellite images created using proprietary data, such as Google software (Google Maps, Street View, and Earth). For more information, see our copyright guidelines: http://journals.plos.org/plosone/s/licenses-and-copyright.

Additional Editor Comments:

The paper present interesting data based in a limited set of SSR to address different questions related to the diversity of continental vs island populations of Planchonella obovata. The reviewers have included some comments in order to improve the manuscript. In my opinion, one major limitation is the number of SSR used in the study, that can limit the demographic models used in the study. Therefore, I suggest to clearly discuss the limitatons of this study and to provide clear information on the validity of these models. I also suggest to reduce the number of figures and tables in the main text, and to avoid including m&M in the introduction (and figures). Please, carefully check the comments to prepare a revised version of your manuscript. Please, also improve the data availiability statement, by including a reference to a public repository or to the data as supplementary material, or other valid option.

Reviewers' comments:

Reviewer's Responses to Questions

**Comments to the Author**

1. Is the manuscript technically sound, and do the data support the conclusions?

Reviewer #1: Yes

Reviewer #2: Yes

2. Has the statistical analysis been performed appropriately and rigorously? 

Reviewer #1: Yes

Reviewer #2: Yes

3. Have the authors made all data underlying the findings in their manuscript fully available?

Reviewer #1: Yes

Reviewer #2: No

4. Is the manuscript presented in an intelligible fashion and written in standard English?

Reviewer #1: Yes

Reviewer #2: Yes

5. Review Comments to the Author

Reviewer #1: In this manuscript, the authors focus on the genetic diversity of plant populations in island environments. Through the study of a tree species, Planchonella obovata, collected on a dozen Japanese islands with contrasting origins (continental vs. oceanic) and ages, they seek to understand the colonisation modalities and to highlight factors contributing to the observed diversity of populations. The present manuscript is well written, with proper analyses and provides interesting results. I only have a reservation about the demographic analyses and estimates of divergence dates. Indeed, the genetic data are limited to 11 SSR markers and the models tested by the ABC approaches are obviously very simple. The authors should be cautious in their interpretation and mention the limitations of these analyses.

Reviewer #2: General comments: This manuscript has scientific merit and significant contribution to population genetics. The objectives are clearly defined. The Materials and Methods section has a strong scientific background. The results are well presented. The discussion shows that the authors have complete understanding on the results, on their implications on genetic diversity, with excellent use of previously published studies. The conclusions are direct related with the objectives. Thus, my recommendation is acceptance with minor revisions. The recommendation of minor revisions is based on some limitations of the current version, emphasized in the following specific comments.

1. Abstract: No materials and methods. Include.

2. Too long introduction (more than three pages including Figure 1). Note that the text in lines 86-87 is objective and the text in lines 118-119 is Materials and Methods. Remove.

3. There is no justification to change the distance to the nearest continent (specify the continent) or the geographical difference between two populations to a natural logarithm scale. Note that the correlation between the geographical difference in km and in Ln(km) is low. I computed 0.14.

4. Is there a justification to Ln-transform the island area?

5. There is no justification to arcsine transformation of He (expected heterozygosity) values. This is unacceptable.

6. It is important to specify the parameter set defined in the Structure software. How you defined the Ancestry model? How you defined the Allele Frequency model?

7. Insert the reference for the DA genetic distance (line 206): Nei et al (1983) J Mol Evol 19:153-170.

8. Figure 2 is referenced in the Materials and Methods section but the content is Result.

9. You can improve the conclusion section simply by providing objective answers to the questions raised in the objectives.

10. There are an excessive number of Figures and Tables. To keep a Table or Figure, it is important that it be informative to the readers. See my opinion below:

Figure 1: Informative; keep.

Table 1: Informative; keep. Please, provide tests for Hardy-Weinberg equilibrium using Fisher’s permutation test or chi-square with Bonferroni correction. In this way, you can assess if there is a significant FIS (inbreeding coefficient) and use this information to discuss population size.

Figure 2: Non-informative; note that you did not reference it in the results section.

Figure 3: Informative; keep.

Figure 4: There is no justification to fit a regression model transforming the geographical distance to a natural logarithm scale. This is unacceptable. Is there a justification to use Fst/(1-Fst)?

Figure 5: I can accept keeping this Figure if the AR and HE were ordered from the lowest value to the highest.

Table 2: Is there a justification to keep?

Figure 6: Informative but only the results for K = 3 should be presented. Remove also the items a) and c). There is no justification to present and, especially, to discuss the results for the other values of the suggested number of clusters. This is also unacceptable. There is no doubt that the Yaeyama and Daito’s plants represent a single population in Hardy-Weinberg equilibrium.

Table 3: Is there a justification to keep?

Table 4: Is there a justification to keep? A Table must be self-informative. None parameters are defined in this table.

6. PLOS authors have the option to publish the peer review history of their article (what does this mean?). If published, this will include your full peer review and any attached files.

Reviewer #1: **Yes: **Philippe LASHERMES

Reviewer #2: No

---

## [Author Response · Author response to Decision Letter 0]

13 Jun 2022

Thank you for your careful consideration to our manuscript. Our answers to reviewers’ comments were followed by the letter “A”.

3. Have the authors made all data underlying the findings in their manuscript fully available?

Reviewer #1: Yes

Reviewer #2: No

A. We have provided data for the SSR primers (accession numbers LC076449- LC076466) and genotypic data of SSRs (S1 Data) used in this study. However, reviewer2 pointed out that we did not provide all data, thus we also provide data used in linear mixed-effect models, examining the associations between genetic diversity and island characteristics (S2 Data), which can be calculated by SSR genotype data and island characteristics information stated in the text (lines 191-192, 730 in the manuscript, and lines 203-204, 764 in the revised manuscript with track changes). 

5. Review Comments to the Author

Reviewer #1: In this manuscript, the authors focus on the genetic diversity of plant populations in island environments. Through the study of a tree species, Planchonella obovata, collected on a dozen Japanese islands with contrasting origins (continental vs. oceanic) and ages, they seek to understand the colonisation modalities and to highlight factors contributing to the observed diversity of populations. The present manuscript is well written, with proper analyses and provides interesting results. I only have a reservation about the demographic analyses and estimates of divergence dates. Indeed, the genetic data are limited to 11 SSR markers and the models tested by the ABC approaches are obviously very simple. The authors should be cautious in their interpretation and mention the limitations of these analyses.

A. Thank you for your comment. We added a sentence explaining the limitation of this analysis in lines 475-478 in the manuscript, and 500-503 in the revised manuscript with track changes, and removed description about divergence time from the conclusion (lines 521-525 in the revised manuscript with track changes).

Reviewer #2: General comments: This manuscript has scientific merit and significant contribution to population genetics. The objectives are clearly defined. The Materials and Methods section has a strong scientific background. The results are well presented. The discussion shows that the authors have complete understanding on the results, on their implications on genetic diversity, with excellent use of previously published studies. The conclusions are direct related with the objectives. Thus, my recommendation is acceptance with minor revisions. The recommendation of minor revisions is based on some limitations of the current version, emphasized in the following specific comments.

1. Abstract: No materials and methods. Include.

A. We added concrete number of samples, populations and genetic markers in the abstract (lines 26-28 in the manuscript and revised manuscript with track changes). More detailed methods would not necessary in the abstract. 

2. Too long introduction (more than three pages including Figure 1). Note that the text in lines 86-87 is objective and the text in lines 118-119 is Materials and Methods. Remove.

A. We deleted the part you mentioned (lines 86-87 and 118-119 in the previous manuscript) and ecological characteristics of the species, which were not necessary in the introduction, were moved to Materials & Methods. We also changed the section title into “Study species and sample collection” from “Sample collection” (lines 121-131 in the manuscript, and lines 133-143 in the revised manuscript with track changes). The introduction length has now decreased to within three pages. 

3. There is no justification to change the distance to the nearest continent (specify the continent) or the geographical difference between two populations to a natural logarithm scale. Note that the correlation between the geographical difference in km and in Ln(km) is low. I computed 0.14.

A. First of all, we have not log-transformed the distance to the nearest continent in in linear mixed-effect models. The nearest continent had been specified as mainland China in the “Statistical analyses of genetic diversity” (line 173 in the manuscript, and line 186 in the revised manuscript with track changes). Transformation of geographical distances between populations and Fst/(1-Fst) (Fig. 5) are recommended in Rousset (1997), and this method is widely adopted. Transformation was made to reduce the skewness and increase the normality of its distribution. We are not sure how obtained the correlation value of 0.14. According to our calculations the correlation coefficient between geographical distance with the closest continent and its logarithmic value was 0.98, and that between inter-population distance and its logarithmic value was 0.86. 

4. Is there a justification to Ln-transform the island area?

A. Transformation of island area is no problem for the same reason as comment 3. We added the explanation in lines 176-177 in the manuscript, and line 189 in the revised manuscript with track changes.

5. There is no justification to arcsine transformation of He (expected heterozygosity) values. This is unacceptable.

A. Transformation of He is no problems for the same reason as comment 3. Especially for He, as its range is from 0 to 1 and like a proportional data, so we selected arcsine transformation. The explanation of transformation had written in lines 183-184 in the manuscript, and lines 195-196 in the revised manuscript with track changes.

6. It is important to specify the parameter set defined in the Structure software. How you defined the Ancestry model? How you defined the Allele Frequency model?

A. We selected "Allele frequency Correlated Model" and "Admixture model" to detect the admixture of lineages. We specified the selected models and reason in lines 199-200 in the manuscript, and lines 211-212 in the revised manuscript with track changes.

7. Insert the reference for the DA genetic distance (line 206): Nei et al (1983) J Mol Evol 19:153-170.

A. We inserted Nei et al. (1983) (ref. no. [46]) in line 212 in the manuscript, and line 224 in the revised manuscript with track changes.

8. Figure 2 is referenced in the Materials and Methods section but the content is Result.

A. We also referenced the Figure 2 in the Results (lines 318, 329, 342, and lines 338, 349, 362 in the revised manuscript with track changes). 

9. You can improve the conclusion section simply by providing objective answers to the questions raised in the objectives.

A. We have rewritten the conclusion providing answers to the question raised in the objectives (lines 481-497 in the manuscript, and lines 506-527 in the revised manuscript with track changes). 

10. There are an excessive number of Figures and Tables. To keep a Table or Figure, it is important that it be informative to the readers. See my opinion below:

Figure 1: Informative; keep.

Table 1: Informative; keep. Please, provide tests for Hardy-Weinberg equilibrium using Fisher’s permutation test or chi-square with Bonferroni correction. In this way, you can assess if there is a significant FIS (inbreeding coefficient) and use this information to discuss population size.

A. All Fis values were not significantly deviated from HWE, and describe the methods and results in lines 149, 170-171 in the manuscript, and lines 161, 181-183 in the revised manuscript with track changes. 

Figure 3: Informative; keep.

Figure 4: There is no justification to fit a regression model transforming the geographical distance to a natural logarithm scale. This is unacceptable. Is there a justification to use Fst/(1-Fst)?

A. Please see answer to comment 3 above.

Figure 5: I can accept keeping this Figure if the AR and HE were ordered from the lowest value to the highest.

A. Four island groups have been ordered from west to east, the same order as Table 1, Figs 2 and 6. We are not sure whether ordering from the lowest value to the highest is really informative. As island age and origin affected the genetic diversity, we reordered then according to the island age, from youngest to oldest, and added the explanation they are ordered according to island age in line 274 in the manuscript, and line 286 in the revised manuscript with track changes.

Table 2: Is there a justification to keep?

A. Table 2 was moved to the supporting information.

Figure 6: Informative but only the results for K = 3 should be presented. Remove also the items a) and c). There is no justification to present and, especially, to discuss the results for the other values of the suggested number of clusters. This is also unacceptable. There is no doubt that the Yaeyama and Daito’s plants represent a single population in Hardy-Weinberg equilibrium.

A. We are afraid to say that we cannot agree with this comment. The �K is not always a good measure of the best K as suggested by Wang (2017) and Funk et al (2020), and we would like to search genetic structure within the well sampled Bonin island group. Thus, we also calculated the best K based on the original method by log-likelihood (Pritchard et al. 2000) and discussed the result from K=3 to K=7. 

Figure 2: Non-informative; note that you did not reference it in the results section.

Table 3: Is there a justification to keep?

A. In order to avoid the confusion of readers about which divergence pattern was the best model, Fig 2 and Table 3 were left in the main text. Moreover, we cited Fig 2 into the Result section. 

Table 4: Is there a justification to keep? A Table must be self-informative. None parameters are defined in this table.

A. Table 4 was moved to the supporting information. We moved a total two tables from the main text. 

References

Funk, S. M., Guedaoura, S., Juras, R., Raziq, A., Landolsi, F., Luís, C., Martínez, A. M., Musa Mayaki, A., Mujica, F., Oom, M. d. M., Ouragh, L., Stranger, Y.-M., Vega-Pla, J. L. & Cothran, E. G. (2020) Major inconsistencies of inferred population genetic structure estimated in a large set of domestic horse breeds using microsatellites. Ecology and Evolution, 10(10), 4261-4279.

Rousset (1997) Genetic differentiation and estimation of gene flow from F-statistics under isolation by distance, DOI: 10.1093/genetics/145.4.1219

Pritchard JK, Stephens M, Donnelly P (2000) Inference of population structure using multilocus genotype data. Genetics 155:945–959

Wang J (2017) The computer program STRUCTURE for assigning individuals to populations: easy to use but easier to misuse. Molecular Ecology Resources 17:981–990

---

## [Editor Report · Decision Letter 1]

17 Aug 2022

Contrasting genetic diversity between Planchonella obovata sensu lato (Sapotaceae) on old continental and young oceanic island populations in Japan

PONE-D-22-03882R1

Dear Dr. Setsuko,

We’re pleased to inform you that your manuscript has been judged scientifically suitable for publication and will be formally accepted for publication once it meets all outstanding technical requirements.

Kind regards,

Judi Hewitt

Academic Editor

PLOS ONE
---

## [Editor Report · Acceptance letter]

23 Aug 2022

PONE-D-22-03882R1 

Contrasting genetic diversity between *Planchonella obovata* sensu lato (Sapotaceae) on old continental and young oceanic island populations in Japan 

Dear Dr. Setsuko:

I'm pleased to inform you that your manuscript has been deemed suitable for publication in PLOS ONE. Congratulations! Your manuscript is now with our production department. 

Kind regards, 

on behalf of

Dr. Judi Hewitt 

Academic Editor

PLOS ONE